# Observational and genetic evidence highlight the association of human sleep behaviors with the incidence of fracture

Yu Qian[1,2,3,7], Jiangwei Xia[1,2,3,7], Ke-Qi Liu[4], Lin Xu[5], Shu-Yang Xie[5], Guo-Bo Chen[6], Pei-Kuan Cong [1,2,3], Saber Khederzadeh[1,2,3] & Hou-Feng Zheng [1,2,3✉]

We combined conventional evidence from longitudinal data in UK Biobank and genetic evidence from Mendelian randomization (MR) approach to infer the causality between sleep behaviors and fracture risk. We found that participants with insomnia showed 6.4% higher risk of fracture (hazard ratio [HR] = 1.064, 95% CI = 1.038–1.090, $P = 7.84 \times 10^{-7}$), falls and bone mineral density (BMD) mediated 24.6% and 10.6% of the intermediary effect; the MR analyses provided the consistent evidence. A U-shape relationship was observed between sleep duration and fracture risk ($P < 0.001$) with the lowest risk at sleeping 7–8 h per day. The excessive daytime sleepiness and "evening" chronotype were associated with fracture risk in observational study, but the association between chronotype and fracture did not show in MR analyses. We further generated a sleep risk score (SRS) with potential risk factors (i.e., insomnia, sleep duration, chronotype, and daytime sleepiness). We found that the risk of fracture increased with an increasing SRS (HR = 1.087, 95% CI = 1.065–1.111, $P = 1.27 \times 10^{-14}$). Moreover, 17.4% of the fracture cases would be removed if all participants exhibited a healthy sleep pattern. In conclusion, insomnia had a causal effect on fracture, falls had a larger intermediary effect than BMD in this association. Individuals with fracture risk could benefit from the intervention on unhealthy sleep pattern.

[1] Diseases & Population Geninfo Lab, School of Life Sciences, Westlake University, 18 Shilongshan Road, Hangzhou, Zhejiang 310024, China. [2] Westlake Laboratory of Life Sciences and Biomedicine, 18 Shilongshan Road, Xihu District, Hangzhou 310024 Zhejiang, China. [3] Institute of Basic Medical Sciences, Westlake Institute for Advanced Study, 18 Shilongshan Road, Hangzhou, Zhejiang 310024, China. [4] WBBC Jiangxi Center, Jiangxi Medical College, Shangrao, Jiangxi, China. [5] WBBC Shandong Center, Binzhou Medical University, Yantai, Shandong, China. [6] Clinical Research Institute, Zhejiang Provincial People's Hospital, People's Hospital of Hangzhou Medical College, Hangzhou, Zhejiang, China. [7] These authors contributed equally: Yu Qian and Jiangwei Xia. ✉email: zhenghoufeng@westlake.edu.cn

Globally, fracture is a primary cause of disability in adulthood, of which fragile fracture is the major type in the elderly[1]. In 2000, the estimated number of new osteoporotic fractures worldwide was 9.0 million for people aged 50 years or more, and Europe accounted for the highest number of fractures (34.8%)[2]. The hip fractures accounted for 18.2% (1.63 million) of all fractures[2], and by 2050, the estimated number of new hip fractures worldwide will increase to 6.26 million cases[3]. According to the report of epidemiology and burden of osteoporosis in the 27 countries of the European Union (EU27), the healthcare cost of fragility fractures was estimated to reach €47 billion for people aged 50 years or more in 2025[4]. Thus, the primary prevention of osteoporotic fractures is important to reduce the cost burden to society.

Both genetic and environmental factors contribute to the risk of fracture[5]. Several genome-wide association studies have identified dozens of genome-wide significant fracture loci[6–8]. All loci were associated with bone mineral density (BMD), suggesting that the lower BMD was the major risk factor for fracture[6]. Besides, environmental factors, such as smoking[9], abdominal obesity[10], and heavy drinking[11], were also found to be associated with the increased fracture risk. Recently, increasing evidence from animal studies has put forward the importance of the circadian system and sleep in the process of bone resorption and formation[12,13]. For example, the clock gene-knockout rats, which cannot regulate the circadian rhythms, displayed abnormal bone mass[13]. Despite these findings, only several observational studies have investigated the relationship between sleep behaviors and fracture risk, but the results were conflicting. For instance, a recent cohort study from the Women's Health Initiative observed that short sleep duration, but not long sleep duration, was associated with increased risk of fracture[14]. However, other studies found that long sleep duration[15], self-reported napping[15], nocturnal hypoxemia[16], and premature awakening[17] were associated with an increased risk of fracture. Besides, another study reported that insomnia was associated with the risk of falls, but not hip fracture[18].

While there were some explanations for the controversial findings, unknown confounding factors in an observational study could be one reason. With the public availability of genetic data for sleep traits, the Mendelian randomization (MR) approach, which is a method that applies genetic variants as instrumental variables (IVs) for the exposure of interest, could strengthen the causal inference on the relationship between exposure and outcome (i.e., fracture)[19], Since the genetic alleles are randomly assorted during conception, MR analyses are less susceptible to confounding factors[19]. Therefore, in this study, we first conducted a prospective observational study to investigate the relationship of insomnia, sleep duration, excessive daytime sleepiness, snoring, and chronotype with fracture risk within the UK Biobank dataset, then, with the summary GWAS data of fracture, we implemented MR approach to detect the causal relationship between these sleep traits and fracture risk. We finally developed a sleep risk score, which integrated the potential fracture-risk sleep traits, and examined the association between sleep risk score and fracture.

## Results

**Insomnia and fracture risk**. After excluding participants with missing data on the main covariates at baseline ($n = 23,688$), 13,002 out of the 398,073 participants (3.27%) developed an incident fracture during a median of 7.96 years of follow-up (Supplementary Table 1). In primary analysis (model 0), an association was observed between insomnia and incident fracture, with a 6.4% higher risk (hazard ratio [HR]: 1.064, 95% confidence interval [95% CI]: 1.038–1.090, $P = 7.84 \times 10^{-7}$) (Fig. 1). Further

adjustments for BMD (model 1) and falls (model 2) attenuated the estimated hazard ratios for the association between insomnia and the incidence of fracture (HR: 1.059, 95% CI: 1.033–1.086, $P = 5.63 \times 10^{-6}$ for model 1; HR: 1.041, 95% CI: 1.016–1.067, $P = 0.001$ for model 2) (Fig. 1). Based on the fully adjusted model (model 3), insomnia was associated with incident fracture, with a 3.7% higher risk (HR: 1.037, 95% CI: 1.012–1.063, $P = 0.004$) (Fig. 1). In addition, we conducted a series of analyses to assess the mediating role of several covariates (i.e., BMD and falls) on the observed associations. The results from the mediation analysis showed that 24.6% and 10.7% of the intermediary effect of insomnia on the risk of fracture was mediated by falls and BMD, respectively (Supplementary Table 2 and Supplementary Table 3). These results suggested that the mediating effect of falls was larger than BMD in the pathway between insomnia and fracture risk. Findings stratified by sex were consistent with the pooled results in model 0 (HR: 1.078, 95% CI: 1.037–1.121, $P = 1.47 \times 10^{-4}$ for male; HR: 1.049, 95% CI: 1.016–1.082, $P = 0.004$ for female) (Fig. 1). Besides, the PAR% for fracture was estimated as 11.3%, suggesting that 11.3% of the fracture cases could be removed if insomnia as a risk factor was removed.

In two-sample MR, genetically predicted insomnia was associated with increased risk of fracture (OR: 1.059, 95% CI: 1.028–1.090, $P = 8.97 \times 10^{-5}$) with the IVW approach. In the weighted median method, the magnitude of the causal association estimate was similar (Fig. 1). MR-Egger regression showed no evidence of directional pleiotropy ($P$ for intercept = 0.711). After excluding one outlier, the causal association estimate was consistently using the MR-PRESSO test (OR: 1.056, 95% CI: 1.027–1.086, $P = 1.94 \times 10^{-4}$) (Fig. 1). The causality between insomnia and the incident fracture was replicated by one-sample MR analysis (HR: 1.061, 95% CI: 1.003–1.123, $P = 0.040$) (Fig. 1). Although evidence from two-sample MR analyses did not support the association between genetically predicted insomnia and BMD (OR: 1.015, 95% CI: 0.995–1.035, $P = 0.146$), there was a causal relationship between insomnia and risk of falls (OR: 1.014, 95% CI: 1.006–1.022, $P = 0.001$), and to some degree supported the hypothesis that the mediating effect of falls was larger than BMD. The bidirectional MR analysis did not suggest the effects of fracture on insomnia (OR: 0.960, 95% CI: 0.896–1.028, $P = 0.244$ for IVW random-effects model).

**Sleep duration and fracture risk**. In this prospective study, there was evidence of a U-shape association between sleep duration and fracture risk in model 0 ($P < 0.001$ for nonlinearity), with the lowest risk of fracture at 7–8 h per day of sleep duration (Fig. 2a, b and Supplementary Fig. 1a–d). Compared with those who slept 7 or 8 h per night, participants who were short (less than 7 h of sleep) and long sleepers (more than 8 h of sleep) had increased risk of fracture in model 0 (HR: 1.154, 95% CI: 1.109–1.202, $P = 2.70 \times 10^{-12}$; HR: 1.099, 95% CI:1.031–1.172, $P = 0.004$, respectively) (Table 1). We found that the effect of short or long sleep duration on fracture risk was attenuated by additionally adjusting for BMD and falls (model 1, model 2, and model 3) (Table 1). Falls and BMD were estimated to mediate 19.0% and 12.7% of the effect of sleep duration on fracture in observational analyses (Supplementary Table 4 and Supplementary Table 5). The results were consistent in the stratified analysis by sex (Supplementary Table 6). Interestingly, compared with participants who slept 7–8 h per night, short and long sleep duration were estimated to explain a similar percentage (5.20%) of the population risk of developing a fracture.

In the two-sample MR, we found that genetically determined increased sleep duration was inversely associated with fracture risk (OR: 0.997, 95% CI: 0.995–0.999, $P = 0.004$) (Table 1). After

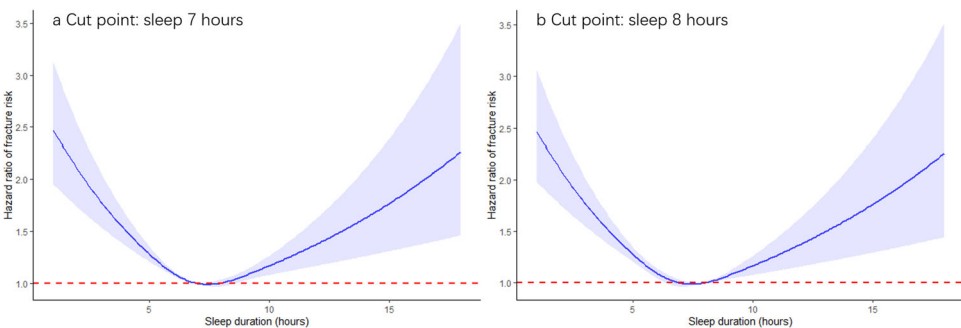

| Method | OR/HR | 95% CI | P value of association |
|---|---|---|---|
| **Observational study** | | | |
| Multivariable Cox regression (model 0) | 1.064 | 1.038-1.090 | $7.84 \times 10^{-7}$ |
| Multivariable Cox regression (model 1) | 1.059 | 1.033-1.086 | $5.63 \times 10^{-6}$ |
| Multivariable Cox regression (model 2) | 1.041 | 1.016-1.067 | 0.001 |
| Multivariable Cox regression (model 3) | 1.037 | 1.012-1.063 | 0.004 |
| One-sample mendelian randomization | 1.061 | 1.003-1.123 | 0.040 |
| **Two-sample mendelian randomization** | | | |
| IVW method | 1.059 | 1.028-1.090 | $8.97 \times 10^{-5}$ |
| Weighted-median method | 1.053 | 1.015-1.093 | 0.006 |
| MR-PRESSO Outlier corrected | 1.056 | 1.027-1.086 | $1.94 \times 10^{-4}$ |
| MR-Egger regression | \ | \ | 0.711* |
| **Observational study** | | | |
| Multivariable Cox regression in male (model 0) | 1.078 | 1.037-1.121 | $1.47 \times 10^{-4}$ |
| Multivariable Cox regression in male (model 1) | 1.074 | 1.032-1.116 | $3.67 \times 10^{-4}$ |
| Multivariable Cox regression in male (model 2) | 1.050 | 1.009-1.091 | 0.015 |
| Multivariable Cox regression in male (model 3) | 1.046 | 1.005-1.087 | 0.026 |
| Multivariable Cox regression in female (model 0) | 1.049 | 1.016-1.082 | 0.004 |
| Multivariable Cox regression in female (model 1) | 1.046 | 1.013-1.080 | 0.006 |
| Multivariable Cox regression in female (model 2) | 1.030 | 0.998-1.064 | 0.068 |
| Multivariable Cox regression in female (model 3) | 1.028 | 0.995-1.061 | 0.094 |

**Fig. 1 Forest plot of observational and Mendelian randomization analyses for the relationships of insomnia with fracture risk.** The 95% confidence interval was presented in the error bar. Model 0 was adjusted for confounders, including age, sex, body mass index, education, smoking, alcohol consumption, physical activity, cognitive impairment, depression, and the use of glucocorticoid medication, benzodiazepines, and antidepressants; Model 1 = Model 0 + BMD; Model 2 = Model 0 + falls; Model 3 = Model 0 + BMD + falls. * the P-value of the intercept term. BMD bone mineral density, CI confidence interval, HR hazard ratio, IVW inverse-variance weighted, MR Mendelian randomization, MR-PRESSO MR pleiotropy residual sum and outlier, OR odds ratio.

**Fig. 2 Observational association of sleep duration with fracture risk using a restricted cubic spline based on model 0 based on different cut points.** **a** Sleeping 7 h per day and **b** sleeping 8 h per day. Hazard ratios are indicated by blue solid lines and the 95% confidence intervals by blue shaded areas. The red line denotes the harzard ratio of one. In all these analyses, models were adjusted for risk factors for fracture, including age, sex, body mass index, education, smoking, alcohol consumption, physical activity, cognitive impairment, depression, and the use of glucocorticoid medication, benzodiazepines, and antidepressants.

excluding one outlier, similar findings were observed in MR-PRESSO test (OR: 0.997, 95% CI: 0.995–0.999, $P = 0.009$) (Table 1). The evidence from the MR-Egger regression also did not support the presence of directional pleiotropy ($P = 0.229$ for intercept). Furthermore, findings from two-sample MR, which genetically predicted sleep duration, were associated with a decreased risk of falls (OR: 0.999, 95% CI: 0.999–0.999, $P < 0.001$), but not with BMD (OR: 1.000, 95% CI: 0.998–1.002, $P = 0.624$). Consistently, the bidirectional MR analyses did not find evidence to support the effect of fracture on sleep duration (OR: 1.000, 95% CI: 0.967–1.034, $P = 0.992$ for IVW random-effect model).

**Snoring, excessive daytime sleepiness, chronotype, and fracture risk**. In the observational study (model 0), we found that snoring was associated with a decreased risk of fracture (HR: 0.951, 95% CI: 0.914–0.989, $P = 0.012$) (Supplementary Table 7). However, while digging into our data, we were aware that participants with snoring were less likely to have insomnia and abnormal sleep duration (Supplementary Table 8). Then, we conducted sensitivity analyses stratified by insomnia symptoms and sleep duration and found the participants free of insomnia and with normal sleep duration, and found that the association between snoring and fracture risk was not statistically significant (HR: 1.063, 95%

CI: 0.965–1.171, $P = 0.218$) (Supplementary Table 7). Similarly, evidence from one-sample MR and two-sample MR did not support this association (OR: 1.214, 95% CI: 0.770–1.913, $P = 0.402$ for two-sample MR; HR: 2.134, 95% CI: 0.932–4.887, $P = 0.073$ for one-sample MR in model 1) (Supplementary Table 7). These results suggested that since participants with snoring were more likely in the low-risk group for other sleep factors (i.e., never/rarely had insomnia and had normal sleep duration), the observational estimates may not reflect the causal effects of snoring on fracture risk.

In multivariable Cox regression (model 0), we found statistically significant associations of daytime sleepiness with the risk of fracture (HR: 1.076, 95% CI: 1.039–1.113, $P = 3.20 \times 10^{-5}$), and being morning preference was a protective factor for fracture risk (HR: 0.963, 95% CI: 0.944–0.982, $P = 1.79 \times 10^{-4}$) (Supplementary Table 9). However, in the two-sample MR, there was no evidence of the association of chronotype (OR: 0.986, 95% CI: 0.954–1.020, $P = 0.425$), which was consistent with results from the one-sample MR (HR: 0.979, 95% CI: 0.916–1.047, $P = 0.538$) (Supplementary Table 9).

**Multivariable MR analyses for sleep behaviors**. In the multivariable MR analyses, where insomnia, sleep duration, snoring,

**Table 1 Observational and Mendelian randomization analyses for the relationships of sleep duration with fracture risk.**

| Method | OR/HR | 95% CI | *P*-value of association |
|---|---|---|---|
| Observational study | | | |
| Multivariable Cox regression (model 0) | | | |
| sleep 7–8 h | 1 (reference) | 1 (reference) | – |
| sleep less 7 h | 1.154 | 1.109–1.202 | $2.70 \times 10^{-12}$ |
| sleep more than 8 h | 1.099 | 1.031–1.172 | 0.004 |
| Multivariable Cox regression (model 1) | | | |
| sleep 7–8 h | 1 (reference) | 1 (reference) | – |
| sleep less 7 h | 1.135 | 1.090–1.182 | $9.79 \times 10^{-10}$ |
| sleep more than 8 h | 1.083 | 1.015–1.155 | 0.016 |
| Multivariable Cox regression (model 2) | | | |
| sleep 7–8 h | 1 (reference) | 1 (reference) | – |
| sleep less 7 h | 1.124 | 1.079–1.170 | $1.47 \times 10^{-8}$ |
| sleep more than 8 h | 1.076 | 1.009–1.148 | 0.025 |
| Multivariable Cox regression (model 3) | | | |
| sleep 7–8 h | 1 (reference) | 1 (reference) | – |
| sleep less 7 h | 1.106 | 1.062–1.152 | $1.16 \times 10^{-6}$ |
| sleep more than 8 h | 1.059 | 0.993–1.130 | 0.081 |
| Two-sample Mendelian randomization | | | |
| IVW method | 0.997 | 0.995–0.999 | 0.004 |
| Weighted-median method | 0.997 | 0.995–0.999 | 0.056 |
| MR-PRESSO Outlier corrected | 0.997 | 0.995–0.999 | 0.009 |
| MR-Egger regression | – | – | 0.229 |

Model 0 was adjusted for confounders, including age, sex, body mass index, education, smoking, alcohol consumption, physical activity, cognitive impairment, depression, and the use of glucocorticoid medication, benzodiazepines, and antidepressants; Model 1 = Model 0 + BMD; Model 2 = Model 0 + falls; Model 3 = Model 0 + BMD + falls. * the *P*-value of the intercept term.
Abbreviations: BMD bone mineral density, CI confidence interval, HR hazard ratio, IVW inverse-variance weighted, MR Mendelian randomization, MR-PRESSO MR pleiotropy residual sum and outlier, OR odds ratio.

and chronotypes were modeled, we found a direct effect of insomnia on fracture risk (OR: 1.055, 95% CI: 1.027–1.085, $P = 1.36 \times 10^{-4}$), and a negative direct effect of sleep duration (OR: 0.997, 95% CI: 0.995–0.999, $P = 0.011$), which is consistent with univariable MR results. These data, taken together, suggested independent causal effects of insomnia and sleep duration on fracture risk.

**The sleep risk score and fracture risk**. Finally, we included four potential risk factors (i.e., insomnia, sleep duration, chronotype, and daytime sleepiness) to develop a sleep risk score (SRS). Each participant received a score of 1 for each of the following sleep behaviors: insomnia ("sometimes" or "usually"), abnormal sleep duration (less than 7 h per day or more than 8 h per day), late chronotype ("evening" or "evening than morning"), and frequent daytime sleepiness ("often" or "all the time") (Table 2). All these component scores were summed to generate an SRS, ranging from 0 to 4. The higher scores indicated poor sleep quality.

By generating the sleep risk score, we assessed the joint effect of sleep behaviors on fracture risk and found that the risk of fracture increased significantly with an increasing SRS (i.e., poor sleep quality) (HR: 1.087, 95% CI: 1.065–1.111, $P = 1.27 \times 10^{-14}$) (Fig. 3). For both vertebral and nonvertebral fracture, such association remained significant (HR: 1.152, 95% CI: 1.011–1.313, *P*-value = 0.033 for vertebral fracture; HR: 1.086, 95% CI: 1.063–1.110, *P*-value = $7.71 \times 10^{-14}$ for nonvertebral fracture). When stratified by sex, the effect estimates of the association between sleep risk score and fracture risk in men were slightly larger than in women (HR: 1.101, 95% CI: 1.065–1.139, $P = 2.01 \times 10^{-8}$ for male; HR: 1.069, 95% CI: 1.040–1.099, $P = 1.88 \times 10^{-6}$ for female) (Fig. 3). Besides, the PAR% for fracture was estimated as 17.4%, suggesting that 17.4% of incident

fracture cases in this study would be removed if all participants had been in healthy sleep behaviors.

Based on the baseline characteristics, participants with healthy sleep quality (i.e., lower sleep risk score) have a decreased risk of falls and higher BMD (Table 2). For example, in the individuals who had SRS = 0, 12.8% of them had falls, compared with 31.3% in SRS = 4 group (Table 2). After adjusting multiple covariates (i.e., model 0), we found that participants with poor sleep quality have an increased risk of falls and reduced BMD (OR: 1.254, 95% CI: 1.241–1.266, *P*-value $< 2 \times 10^{-16}$ for falls, and β-coefficient = −0.004, SE = $2.59 \times 10^{-4}$, *P*-value $< 2 \times 10^{-16}$ for BMD). We then further included BMD and falls as mediators in the adjusted model and found that the magnitude of the associations of SRS with fracture risk was attenuated (HR: 1.054, 95% CI: 1.031–1.077, $P = 2.24 \times 10^{-6}$ for model 3) (Fig. 3). The mediation analyses showed that 18.2% and 12.2% of the intermediary effect of SRS on the risk of fracture was mediated by falls and BMD (Supplementary Table 10 and Supplementary Table 11).

## Discussion

In this study, using a large-scale UK Biobank dataset, we investigated the association of five sleep behaviors (insomnia, sleep duration, snoring, daytime sleepiness, and chronotype) with fracture risk. Our findings suggest that unhealthy sleep patterns could result in a higher risk of fracture, and 17.4% of all fracture cases could be removed if all participants exhibited healthy sleep patterns. To some extent, falls had a larger intermediary effect than BMD in the association between sleep and fracture. Both the observational study and MR analyses consistently suggested a causal effect of insomnia on the fracture risk. Further, a U-shape relationship was observed between sleep duration and fracture risk, with the lowest risk of fracture at 7–8 h per day of sleep duration.

Previous observational studies examining the associations between insomnia and fracture risk were few, and existing evidence was inconsistent. For example, one study that included 34,163 nursing home residents found no association between insomnia and the risk of hip fracture (OR: 0.99, 95% CI: 0.77–1.26)[18]. However, due to the limited validity of the insomnia Minimum Data Set items used in the analysis, it is possible that their conclusion was prone to bias[20]. In contrast, the results from another cohort study in the Women's Health Initiative (WHI) suggested an association between insomnia and an elevated risk of total fracture (HR: 1.03, 95% CI: 1.01–1.06) in multivariable models[14], which agreed with our findings. Furthermore, using genetic variants associated with insomnia phenotype, we found strong evidence for the causal effect of insomnia on fracture risk.

Recent epidemiological studies have reported a J-shaped or reverse J-shaped association between sleep duration and fracture risk, and the evidence is not consistent. For example, a cohort study including 157,306 women found that, compared with subjects sleeping 7 h per night, short sleepers (≤5 h per night) had a 12% (95% CI: 5–20%) increased odds of all fractures, but no association was found between fracture risk and a long duration of sleep[14]. In contrast, in another population-based cohort study that included 8101 women, a long duration of sleep (but not short sleep time) was associated with an increased risk of nonspinal fractures (HR: 1.26, 95% CI: 1.00–1.58 for sleeping ≥10 h)[15]. In our study, the evidence from observational study suggested a U-shape association between sleep duration and fracture risk. Participants with a sleep duration of 7–8 h per day had the lowest fracture risk, suggesting that sleeping 7–8 h per day might be the appropriate sleep duration for preventing fracture. However, findings from two-sample MR supported a linear adverse effect of sleep duration (continuous trait as exposure) on the fracture risk.

**Table 2 Baseline characteristics of participants with different sleep behaviors in prospective studies.**

| Baseline characteristics | Sleep risk score | | | | |
|---|---|---|---|---|---|
| | 0 | 1 | 2 | 3 | 4 |
| Number of participants | 50331 | 153801 | 110621 | 31843 | 2596 |
| Having low-risk sleep behaviors[b] | | | | | |
| Never/rarely insomnia | 50331 (100.0) | 27275 (17.7) | 7063 (6.4) | 633 (2.0) | 0 (0.0) |
| Sleep 7–8 h per day | 50331 (100.0) | 140979 (91.7) | 47309 (42.8) | 3103 (9.7) | 0 (0.0) |
| Morning perference | 50331 (100.0) | 149145 (97.0) | 96874 (87.6) | 21907 (68.8) | 0 (0.0) |
| Never/rarely dozing | 50331 (100.0) | 144004 (93.6) | 69996 (63.3) | 6200 (19.5) | 0 (0.0) |
| Bone mineral density[a] | 0.56 (0.13) | 0.54 (0.13) | 0.54 (0.14) | 0.54 (0.14) | 0.54 (0.15) |
| Falls[b] | | | | | |
| No | 43905 (87.2) | 127894 (83.2) | 87656 (79.2) | 23463 (73.7) | 1783 (68.7) |
| Yes | 6426 (12.8) | 25907 (16.8) | 22965 (20.8) | 8380 (26.3) | 813 (31.3) |
| Fracture[b] | | | | | |
| No | 48986 (97.3) | 148945 (96.8) | 106753 (96.5) | 30615 (96.1) | 2479 (95.5) |
| Yes | 1345 (2.7) | 4856 (3.2) | 3868 (3.5) | 1228 (3.9) | 117 (4.5) |

[a]Values are mean (SD).
[b]Values are numbers (percentages).

**Fig. 3 Forest plot of observational analyses for the relationships of sleep risk score with fracture risk.** The 95% confidence interval was presented in the error bar. Model 0 was adjusted for confounders, including age, education, sex, smoking, alcohol consumption, physical activity, body mass index, and the use of glucocorticoid, benzodiazepines, and antidepressants; Model 1 = Model 0 + BMD; Model 2 = Model 0 + falls; Model 3 = Model 0 + BMD + falls. BMD bone mineral density, CI confidence interval, IVW inverse-variance weighted, MR Mendelian randomization, MR-PRESSO MR pleiotropy residual sum and outlier, OR odds ratio.

Due to the limited data[21], we did not conduct the MR analyses while taking long/short duration (binary trait) as exposures. We found that, daytime sleepiness was associated with risk of fracture, but being morning preference was a protective factor for fracture risk in observational study. However, our MR analyses found no significant association between genetically determined chronotype and the incidence of fracture, but we cannot preclude weak genetic associations because of the limited statistical power in this study (i.e., 15.4%).

Furthermore, for snoring, a cohort study including 3220 women and 2699 men found no significant association between snoring with a high fracture risk. However, female participants in the severe-snoring group (6–7 nights per week or sleep disturbance by snoring in the next room) had an increased risk of fracture (HR: 1.682, 95% CI: 1.164–2.430, $P = 0.006$)[22]. Similarly, a prospective study that included 55,264 women in the Nurses' Health Study (NHS) also found that participants with a history of obstructive sleep apnea, a sleep disorder associated with snoring[23], were at a higher risk of vertebral fracture (HR: 1.88, 95% CI: 1.21–2.93)[24]. In our observational study, snoring participants rarely had insomnia and abnormal sleep duration. After excluding participants with insomnia or short/sleep duration, we did not find a significant association between snoring and the risk of any fracture. MR analysis results also did not support a causal role of snoring. Taken together, these data suggested that snoring may not be a significant modifiable risk factor for fracture. However, due to the limited data on snoring (e.g., the frequency

of snoring), we could not assess the association between severe snoring and fracture risk.

Because sleep behaviors might affect each other[25], for example, late chronotype and insomnia might result in shorter sleep duration and excessive daytime sleepiness, these sleep behaviors might individually act through different mechanisms, but could work synergistically to increase the risk of fracture. For instance, insomnia and abnormal sleep duration were relevant to the metabolic disruption (e.g., decreased melatonin secretion)[26], and activated inflammation[27], which might increase the risk of fracture[28,29]. Late chronotype had been linked to disrupted circadian rhythm[30,31], and the deficiency of clock gene (e.g., BMAL1) in mice could result in a low bone mass[32]. Additionally, circadian disruption could also increase sleepiness and possibly adversely affect vigilance to environmental hazards[33], which result in the increased risks of falls and fractures[15]. Therefore, we constructed a SRS to assess the overall relationship between the combination of sleep behaviors and fracture risk. We found that poor sleep quality was associated with an increased risk of fracture, and 17.4% of the fracture cases in this population would be removed if all participants exhibited a healthy sleep pattern. These data suggested the importance of careful consideration of sleep patterns, particularly insomnia, in fracture-risk assessment and prevention in clinical practice.

Given the associations of sleep disturbance with low bone mineral density (BMD)[34] and the loss of postural control[35], the potential relationship between sleep behavior and fracture risk

might be caused by the mediating effect of either BMD-related or fall-related factors. In our adjusted models, when including BMD and falls as additional covariables, the magnitude of the associations between sleep behaviors and fracture risk would attenuate, with wider confidence intervals. We observed that a larger proportion of the effect of sleep on fracture risk was mediated by falls than by BMD (e.g., 24.6% vs 10.7% for insomnia). Also, estimates from MR analyses supported the causal relationship between several sleep behaviors (e.g., insomnia or sleep duration) and the risk of falls. Taken together, these results suggested the stronger mediating role of falls in the association between sleep and fracture risk.

Despite the comprehensive methodologies used in the analyses, this study had a few limitations. The population included in this study was exclusively European and of a modest age range (37–73 years), which may limit the generalizability of our findings to other races (e.g., Asian) and younger populations. Additionally, due to the sample overlap between the datasets used in the one-sample and two-sample MR analyses (e.g., snoring and sleep duration), one-sample MR results could not considered as independent replication of the two-sample MR results.

In summary, our observational and Mendelian randomization findings supported an association between poor sleep quality and an increased risk of fracture. The sleep risk score defined by our study (evening preference, sometimes or usually insomnia, abnormal sleep duration, and often or all-the-time daytime sleepiness) provided a frame of reference for identifying individuals at high risk of fracture. Among the five sleep behaviors, insomnia had a causal effect on fracture risk. Furthermore, fall prevention for reducing fracture risk in individuals with poor sleep quality could be more efficient than improving BMD.

## Methods

### Study design and data sources

*Individual-level data.* The overall study design is presented in Fig. 4. Briefly, we employed the individual-level data from the UK Biobank (Application 41376) as we used before[36,37]. In the UK Biobank dataset, information on more than 2000 traits, including sleep behaviors, fractures, and relevant confounding factors, was recorded through questionnaires and physical measurements. Participants were genotyped with UK Biobank Axiom Array, and genotype imputation was performed using the 1000 Genomes Project (Phase 3) reference panel[38]. Ethics approval for the UK Biobank research was obtained from the North West Multicentre Research Ethical Committee, and all participants provided informed consent. This study was performed under generic ethical approval obtained by UK Biobank from the National Health Service National Research Ethics Service (approval letter ref 12/NW/0382, 17 June 2011).

We excluded 30,486 non-European participants to minimize the population-stratification bias. Then, we excluded participants with potential comorbid diseases (i.e., rheumatoid arthritis, ulcerative colitis, multiple sclerosis, Crohn's disease, hyperthyroidism, or lupus erythematosus) ($N = 28,657$). Additionally, since this study is a prospective design, we also excluded participants who had suffered a fracture before baseline, and participants with secondary fractures (i.e., participants with follow-up care involving removal of the fracture plate and other internal fixation devices, pathological fractures, and the fracture of the bone in neoplastic disease), leaving 421,761 participants in the observational study. In the one-sample MR analysis, we further excluded 139,599 participants (Field ID 22021) who had a kinship with other participants. Supplementary Data 1 listed the field ID and code for participants who were excluded in quality control.

*Summary-level data and selection of instrumental variables.* Single-nucleotide polymorphisms (SNPs) associated with the respective sleep behavior (chronotype[39], insomnia[40], sleep duration[21], or snoring[41]) at a genome-wide significant ($P < 5 \times 10^{-8}$) level were selected from the largest genome-wide association studies ($n = 697,828$ for chronotype; $n = 1,331,010$ for insomnia; $n = 446,118$ for sleep duration; $n = 408,317$ for snoring) (Supplementary Table 12). To reduce the influence of linkage disequilibrium (LD) on MR analysis[42], we extracted SNPs with the lowest $P$-value for the associated trait after clumping for LD at $r^2 < 0.01$ in 250-kb regions. As a result, 268 SNPs were retained as instrumental variables for chronotype, 184 SNPs for insomnia, 72 SNPs for sleep duration, and 30 SNPs for snoring (Supplementary Data 2). By dividing the square of the SNP-exposure association estimate by the square of the corresponding SNP-exposure standard error[43], we calculated the F-statistic for the SNP set used as instrumental variables for sleep-associated traits. We found that all instrumental variables used greatly

exceed the criteria of strong instruments (F-statistic > 10), ranging from 26.9 to 220.8. Summary-level data for fracture (the outcome) were available from the GWAS that included 53,184 cases and 373,611 controls[7]. Additionally, genetic association estimates for BMD were obtained from a GWAS of 426,824 individuals[7]. We further used summary statistics for falls from a GWAS, including 361,194 participants of European ancestry (http://www.nealelis/uk-biobank/). For SNPs that were not available in the outcome datasets, we used proxy SNPs where available (i.e., genetic variants in LD with the corresponding SNPs at the threshold of $r^2 > 0.9$) (Supplementary Data 2). In the bidirectional MR analyses, we had a total of 14 genetic variants as instruments for fracture, after clumping for linkage disequilibrium at $r^2 < 0.01$ in 250-kb regions that reached genome-wide significance ($P < 5 \times 10^{-8}$)[7].

### Assessment of sleep behaviors in UK biobank and definition of sleep risk score. In the UK Biobank, participants reported five sleep behaviors: insomnia, sleep duration, snoring, daytime sleepiness, and chronotype (Supplementary Data 3). Insomnia symptoms were assessed in the question, "Do you have trouble falling asleep at night or do you wake up in the middle of the night?" with the three possible answers being "never/rarely", "sometimes", or "usually" or "prefer not to answer", which we coded as 0, 1, and 2, respectively. Sleep duration was coded based on the number of reported hours using the following question: "About how many hours of sleep do you get in every 24 h? (including naps)". Information on snoring was obtained by asking, "Does your partner or a close relative or friend complain about your snoring?" with responses being either "no" or "yes". We derived a binary variable for snoring where "no self-reported snoring" and "having self-reported snoring" were coded as 0 and 1, respectively. To assess subjective daytime sleepiness, participants were asked, "How likely are you to doze off or fall asleep during the daytime when you don't mean to? (e.g., when working, reading, or driving)" with responses being "never/rarely", "sometimes", "often", or "all the time". We derived an ordinal variable for subjective daytime sleepiness where "never/rarely", "sometimes", "often", and "all the time" were coded as 0, 1, 2, and 3, respectively. Information on chronotype preference was collected by asking, "Do you consider yourself to be (i) definitely a 'morning' person, (ii) more of a 'morning' than an 'evening' person, (iii) more of an 'evening' than a 'morning' person, (iv) definitely an 'evening' person, (v) do not know, or (vi) prefer not to answer"? We derived a four-level ordinal variable for chronotype preference where "an 'evening' person", "more of an 'evening' than a 'morning' person", "more of a 'morning' than an 'evening' person", and "a 'morning' person" were coded as 0, 1, 2, 3, respectively. For the above questions, participants who responded "Prefer not to answer" or "Do not know" were set to missing. We included the potential fracture-risk sleep factors to generate a sleep risk score (SRS). For each high-risk sleep behavior, the participant received a score of 1 if he or she met the criterion for high risk (see Results). If the participant did not meet the criterion, he or she was classified at low risk for that behavior, and received a score of 0. SRS was calculated as the sum of all these component scores, with higher scores indicating poor sleep quality.

### Assessment of fracture in UK biobank. In this prospective study, the endpoint was the first diagnosis of any fracture (except secondary fracture). The fracture could be located at the skull and face, neck, vertebrae, ribs, sternum, and thoracic spine, shoulder and upper arm, spine, forearm, wrist, and hand, pelvic and thigh, femur, lower leg, foot except for the ankle, and other unspecified body regions. These cases were included from either questionnaire-based self-reported fractures or hospital-based fractures using the ICD-9 or ICD-10 codes (Supplementary Data 3).

### Statistics and reproducibility

*Observational study.* In the prospective study of fracture risk, follow-up time was calculated from the date of attending the UK Biobank to the diagnosis of fracture, death, or the censoring date (31 March 2017). Five sleep behaviors (chronotype, insomnia, snoring, sleep duration, and excessive daytime sleepiness) and sleep risk scores were assessed in relation to the fracture risk using Cox regression. In the multivariable Cox regression, the basic model was adjusted for confounders, including age, sex, body mass index, education, smoking, alcohol consumption, physical activity, cognitive impairment, depression, and the use of glucocorticoid medication, benzodiazepines, and antidepressants (model 0); additional covariables such as BMD and falls were also included to set different models (model 1 = model 0 + BMD, model 2 = model 0 + falls, and model 3 = model 0 + BMD + falls). Detailed information on the corresponding covariates was provided in Supplementary Data 4.

Restricted cubic spline with five knots at 5th, 35th, 50th, 65th, and 95th centiles was used to explore the potential nonlinear association of sleep duration with fracture risk. Assuming a causal relationship, we calculated proportional population-attributable risk (PAR%) to estimate the proportion of the fracture incidence in this population that would be eliminated if the exposure were eliminated. We then conducted mediation analyses to estimate the dimensionless proportion of the effect of sleep behaviors on fracture risk mediated by BMD and falls.

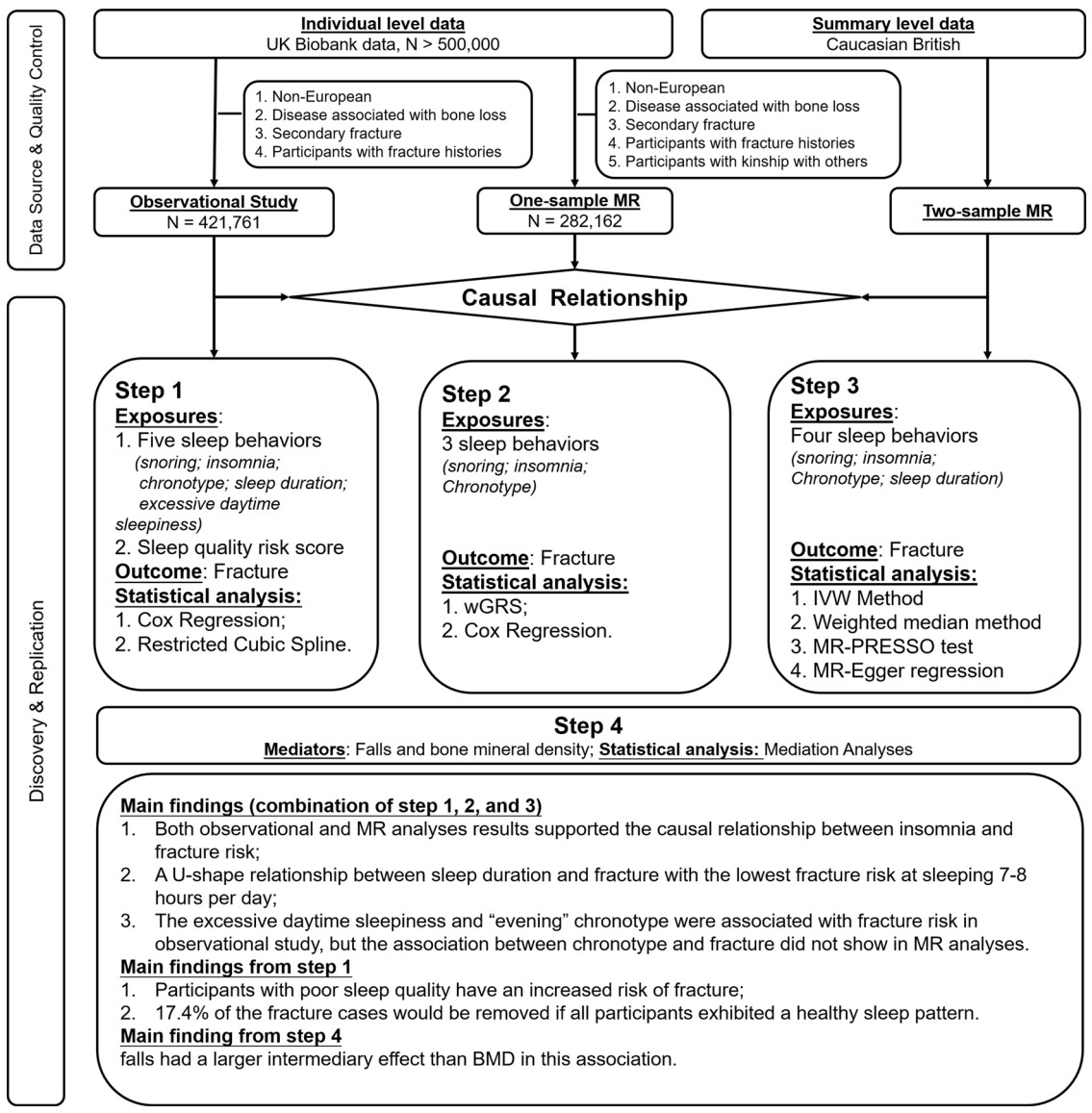

**Fig. 4 An overview of the study design.** IVW inverse-variance weighted, MR Mendelian randomization, MR-PRESSO MR pleiotropy residual sum and outlier, wGRS weighted genetic risk score.

To account for multiple comparisons, a Bonferroni-corrected threshold of $P < 0.01$ (0.05/5 sleep behaviors) was considered to be statistically significant. Findings with $P$-values less than 0.05 but above the threshold of Bonferroni-corrected significance were considered to be suggestive evidence. All these statistical analyses were conducted in R version 4.0.2 (http://www.r-project.org/), and the PLINK2.0 software[44].

*One-sample MR study.* In one-sample MR analyses, after extracting SNPs from the UK Biobank imputation dataset (Supplementary Data 2), weighted genetic risk score (wGRS) for four sleep behaviors (chronotype, insomnia, snoring, and sleep duration) was generated using the following equation (Eq. 1):

$$\text{wGRS} = \sum_{i=1}^{n} \beta_n \times \text{SNP}_n, \qquad (1)$$

where $\text{SNP}_n$ was the dosage of the effective allele, $\beta_n$ was the effect estimate of $\text{SNP}_n$ for each sleep behavior obtained from the previous GWAS, and $n$ was the number of instrumental variables ($n = 268$ for chronotype, 184 for insomnia, 72 for sleep duration, and 31 for snoring) (Supplementary Data 2). The wGRS method was performed using the PLNK2.0 software with the command of -score sum[44].

After obtaining the sleep-behavior wGRS, we performed a multivariable Cox regression to obtain the population average causal hazard ratio using the prospective study of fracture risk, based on model 0.

*Two-sample MR study.* After extracting the genetic association estimates of the outcomes from summary-level data, ratio estimates for individual SNPs were

calculated using the Wald estimator and Delta method[45]. Due to the presence of heterogeneity for each outcome (all $P$ values for Cochran's Q test < 0.05), the inverse-variance-weighted (IVW) method (under random-effect models) was further used to combine these estimates, thus obtaining the primary causal estimates[45]. In sensitivity analyses, to assess the influence of potential pleiotropy on the causal effect estimates, the weighted-median method, MR Pleiotropy Residual Sum and Outlier (MR-PRESSO) test, and MR-Egger regression were conducted. Specifically, the weighted median method can provide valid estimates if more than 50% of the weights come from valid instrumental variables[46]. Additionally, the MR-PRESSO test can detect and correct for the influence of outliers on MR estimates[47], and in MR-Egger regression, the $P$-value of the intercept term can be used as an indicator of directional pleiotropy ($P$-values less than 0.05 were considered statistically significant)[48]. To address the direction of causality, we further performed the bidirectional, after extracting the genetic association estimates for the sleep behaviors and fractures from GWASs of the corresponding phenotype[7,21,39–41].

*Multivariable MR analysis.* Using the MendelianRandomization package, we conducted the multivariable MR analyses to find the independent effect of sleep behaviors on fracture risk. Briefly, we constructed instrumental variables by the combination of SNPs and their genetic association estimates from the GWASs for the sleep behaviors[21,39–41]. The genetic association estimates of these instrumental variables with other exposures (i.e., other sleep-related factors, except for the originally associated sleep behavior) and outcome (i.e., fracture) were obtained from the summary statistics from GWAS in the UK Biobank[7,40].

**Reporting summary**. Further information on research design is available in the Nature Research Reporting Summary linked to this article.

## Data availability

The source values for Figs. 1 and 3 have been provided in Supplementary Data 5. The genetic summary statistics for BMD and fracture can be obtained from http://www.gefos.org/?q=content/data-release-2018, and the summary-level data for falls can be downloaded from www.nealelab.is/uk-biobank/. The individual-level genetic and phenotype data require the permission from the UK Biobank.

## Code availability

The code is available from the corresponding author upon reasonable request.

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

## Acknowledgements

This work was supported by the National Natural Science Foundation of China (Grant Nos: 81871831 and 32061143019). We thank the GEnetic Factors for OSteoporosis (GEFOS) Consortium and the UK Biobank (Application 41376). We also thank the High-Performance Computing Center at Westlake University for the facility support and technical assistance.

## Author contributions

H.-F.Z. conceptualized and designed the study. Y.Q. and J.X. conducted analysis. Y.Q. drafted the paper, P.-K.C., G.-B.C., K.-Q.L., L.X., S.-Y.X. and S.K. reviewed and edited paper. All authors contributed, discussed and approved paper.

## Competing interests

The authors declare no competing interests.
