## [Peer Review File · Communications Biology]

Reviewers' comments:

Reviewer #1 (Remarks to the Author):

Overall Consideration

The paper represents a prospective study of the association between sleep problems and fractures. The authors conclude that insomnia increased the risk of fracture, mediated by falls and BMD. However, independence of the association with the fracture may very well be due to confounding as residual confounding is likely to present, e.g. antidepressant use, comorbidities etc. The basic associations lack novelty as there are a number of relevant papers which have very similar findings based on longitudinal studies; overall this study is not an addition to the literature. Although the mendelian randomization (MR) approach was used in this study suggesting a causality, previous studies based on longitudinal data also can infer a causal relationship. Importantly, conclusions with respect to fracture risk are over-stated. I hope the authors find some specific comments below helpful in improving the manuscript.

Subgroup analysis by sex was performed in this study, its rationale should be stated.

Mediation analysis was also performed for falls and BMD, its rationale needs to be stated as well. If the authors consider they are mediators, what are the effects of sleep on falls and BMD in this study?

I am concerned that comorbid conditions/poorer health, depression/depressive symptoms and poor cognitive function may be the reason for sleep disturbance and fractures. There may be residual confounding.

Additionally, certain medications (e.g. benzodiazepines, antidepressants) may increase sleep disturbance and fractures.

It's known that the mechanisms/risk factors for the site of fracture (e.g. vertebral vs. non-vertebral) are different, grouping fracture by the site can tell this difference and also make your study comparable with previous studies.

Did the authors adjust for baseline fracture? Those with fractures at baseline are more likely to have fractures in the future.

Discussion about why the results were different with different sleep behaviours is needed.

Discussion about potential mechanisms for sleep-fracture associations is also needed.

Reviewer #2 (Remarks to the Author):

The authors conducted a prospective study to infer the causality between sleep behaviors and fracture risk. Thanks to all the authors for this interesting manuscript in this important area. I suggest a minor revision for this manuscript. The comments are shown as follows:

1. Title

Please add the type of article in the title, I suggest "Observational and genetic evidence on the association of sleep behaviors with the incidence of fracture: A prospective study."

2. Abstract

Line 17: Please note the whole worlds of BMD for the first time in the manuscript;

3. DISCUSSION

Please clearly describe the LIMITATION of this study in the discussion.

4. OVERALL

The paper presents a topic of interest to the researchers in the related areas but the paper needs some language improvement before acceptance for publication.

Reviewer #3 (Remarks to the Author):

In this publication the authors triangulated various sources of statistical evidence to investigate the relationship between sleep traits and fracture risk, and concluded that an association is likely to exist between poor sleep quality and fracture risk. I enjoyed reading this paper- I found the research question interesting and the manuscript to be well written. I'm not an expert on statistical analysis involving longitudinal data so cannot comment on this in detail but I am well acquainted with two sample MR analysis. For these sections of the paper, the analysis was well conducted, and I support the conclusions that the authors make. I do however have some comments for improvements:

Major comments

Methods

- What p-value cut-off was used to select the instruments for the sleep traits, and how was the LD clumping conducted to ensure the instruments were independent? This detail is provided in the supplementary methods, but I think it is important and therefore would be better placed in the main manuscript.
- For the same reason, I think details of the GWASs used for the exposures and outcomes tested in the two sample MR, which are currently reported in supplementary table 2, should be provided in the main text with citations to the relevant publications.
- My understanding for the two sample MR study is that the authors used the sleep trait GWAS data from UK BioBank and then linked this to fracture risk measured in another study. I think clarification is needed on the sample overlap between the datasets used in the one-sample MR and two-sample MR study designs, and therefore whether these findings can be interpreted as replication from independent data sources or not.
- As there is a lot of different results presented in this paper, I think the reader would benefit from having the flow chart in supplementary figure 1 extended to summarize the main findings from each part of the study.

Results

- F-statistics should be derived and reported to evaluate instrument strength for all the MR analyses.
- Q-statistics should also be derived and reported to evaluate instrument heterogeneity.
- I think it would be informative to conduct a two sample MR-PheWas between the sleep trait instruments across fracture and bone-related risk factors and disease outcomes to see if any evidence of vertical pleiotropy can be detected, which would further support the causal relationship observed with fracture risk.
- Steiger analysis or bi-directional MR should be performed to confirm correct orientation of causal effect.

Minor comments

These are recommendations for further analyses that might be useful and strengthen the conclusions of the paper. However, each analysis may require a lot of work so I don't think necessary to put in the manuscript, but the authors may want to run and include some of these.

- Any fracture was used as an outcome in this study. Would it be possible to repeat analyses using fracture data for the different sites as outcomes, and compare findings? A fracture site specific

analysis might provide better power to detect sleep related effects and potentially help to further understand the causal mechanism (could be a discussion point in the manuscript).

- Including multivariable MR analysis (MVMR) as follow-up for the two sample MR, where the sleep trait exposures are modelled together to find the independent MR effects on risk fracture could be informative.
- For completeness, it might be worthwhile conducting an MR analysis between your sleep risk score and fracture risk.

Title: Observational and genetic evidence on the association of sleep behaviors with the incidence of fracture

Thank you for the opportunity to resubmit our manuscript to *Communications Biology* for consideration for publication. For ease of reading we have directly pasted the Reviewer's comments below, which are in Calibri font.

Our responses are in Times New Roman font, preceded by “**Response:**”

Reviewers' comments:

Reviewer #1 (Remarks to the Author):

Overall Consideration

The paper represents a prospective study of the association between sleep problems and fractures. The authors conclude that insomnia increased the risk of fracture, mediated by falls and BMD. However, independence of the association with the fracture may very well be due to confounding as residual confounding is likely to present, e.g. antidepressant use, comorbidities etc. The basic associations lack novelty as there are a number of relevant papers which have very similar findings based on longitudinal studies; overall this study is not an addition to the literature. Although the mendelian randomization (MR) approach was used in this study suggesting a causality, previous studies based on longitudinal data also can infer a causal relationship. Importantly, conclusions with respect to fracture risk are over-stated. I hope the authors find some specific comments below helpful in improving the manuscript.

Subgroup analysis by sex was performed in this study, its rationale should be stated.

Response: Osteoporosis was thought of as a “woman’s disease” because the prevalence of osteoporosis is much higher in postmenopausal women than in older men.(1) Despite the higher fracture risk in postmenopausal women, older men tend to have worse outcomes after a fracture. Males and females might have different disease mechanisms.(1) Therefore, to avoid the bias of residual confounding of gender, we also conducted the subgroup analyses by sex. We found that poor sleep quality is a risk factor for both males and females.

Mediation analysis was also performed for falls and BMD, its rationale needs to be stated as well. If the authors consider they are mediators, what are the effects of sleep on falls and BMD in this study?

Response: Thank you for your suggestion. Low BMD was considered as the most common risk factor for fracture.(2) Additionally, sleep disturbances could also reduce vigilance to environmental

hazards, and had a potential adverse effect on balance,(3) causing an increased risk of falls.(4) Therefore, in the multivariable Cox regression, the basic model was adjusted for confounders, including age, sex, body mass index, education, smoking, alcohol consumption, physical activity, cognitive impairment, depression, and the use of glucocorticoid medication, benzodiazepines, and antidepressants (model 0), additional covariables such as BMD and falls were also included to set different models (model1=model 0+BMD, model 2=model0+falls, and model 3=model 0+BMD+falls). In our observational analyses, linear regression was used to assess the association between sleep risk scores and BMD, and logistic regression was used to determine the association for falls.

As you suggested, we evaluated the effects of sleep on falls and BMD. After adjusting multiple covariates (i.e., age, sex, body mass index, education, smoking, alcohol consumption, physical activity, cognitive impairment, depression, and the use of glucocorticoid medication, benzodiazepines, and antidepressants), we found that participants with higher sleep risk scores (i.e., poor sleep quality) were more likely to develop falls and lower BMD (OR: 1.254, 95% CI: 1.241-1.266, P -value $< 2 \times 10^{-16}$ for falls, and β -coefficient = -0.004, SE = 2.59×10^{-4} , P -value $< 2 \times 10^{-16}$ for BMD). Additionally, results from mediation analyses showed that the mediating effect of falls was larger than BMD in the pathway between sleep quality and fracture risk. We have added the results of these analyses in the Result section on Page 15, Lines 390-393, which were highlighted in red.

I am concerned that comorbid conditions/poorer health, depression/depressive symptoms, and poor cognitive function may be the reason for sleep disturbance and fractures. There may be residual confounding. Additionally, certain medications (e.g. benzodiazepines, antidepressants) may increase sleep disturbance and fractures.

Response: Thank you for your suggestions. In our observational analyses, we first excluded participants with potential comorbid diseases (i.e., rheumatoid arthritis, ulcerative colitis, multiple sclerosis, Crohn's disease, hyperthyroidism, or lupus erythematosus) and participants with secondary fractures (i.e., participants with follow-up care involving removal of the fracture plate and other internal fixation devices, pathological fractures, and the fracture of the bone in neoplastic disease) (Supplementary Table 1 or table below).

Additionally, as you suggested, we further included cognitive impairment, depression, and the use of glucocorticoid, benzodiazepines, and antidepressants in our multivariable Cox proportional hazards model. The hazard ratio (HR) with 95% confidence intervals (CIs) for the risk of disease was progressively adjusted for age, sex, BMI, smoking, alcohol consumption, physical activity, education, cognitive impairment, depression, and the use of glucocorticoid, benzodiazepines, and antidepressants (Supplementary Table 5 or table below). We found that participants with poor sleep quality increased the risk of fracture (HR: 1.064, 95% CI: 1.038-1.090, $P = 7.84 \times 10^{-7}$ for insomnia; HR: 1.154, 95% CI: 1.109-1.202, $P = 2.70 \times 10^{-12}$ for less than 7 hours of sleep; HR: 1.099, 95% CI: 1.031-1.172, $P = 0.004$ for more than 8 hours of sleep; HR: 1.076, 95% CI: 1.039-1.113, $P = 3.20 \times 10^{-5}$ for daytime sleepiness; HR: 1.076, 95% CI: 1.039-1.113, $P = 3.20 \times 10^{-5}$ for chronotype; HR: 1.087, 95% CI: 1.065-1.111, $P = 1.27 \times 10^{-14}$ for sleep risk score). We have updated the results

in Abstract and the Result section on Page 10-15, we also revised the Method section (Lines 104-114 on Page 5 and Lines 191-193 on Page 8). The changes were highlighted in red.

Detailed information on the field ID and code for participants who were excluded in sample quality control and covariates in UK Biobank.

Phenotype	Field ID	Code
Fracture of bone in neoplastic disease	41270	M907, M9070, M9071, M9072, M9073, M9074, M9075, M9076, M9077, M9078, M9079
Follow-up care involving removal of fracture plate and other internal fixation devices	41270	Z470
Pathological fracture	41203	7331, 73313, 73315, 73316
	41270	M800, M8000, M8001, M8003, M8007, M8008, M801, M8010, M802, M8021, M8028, M804, M8040, M8041, M8046, M8048, M8049, M805, M8050, M8052, M8053, M8055, M8058, M808, M8080, M8081, M8082, M8086, M8088, M8089, M809, M8090, M8091, M8092, M8093, M8094, M8095, M8096, M8097, M8098, M8099; M844, M8440, M8441, M8442, M8443, M8444, M8445, M8446, M8447, M8448, M8449
Rheumatoid arthritis	20002	1464
	41270	M053, M0530, M0538, M0539, M058, M0580, M0582, M0583, M0584, M0586, M0587, M0588, M0589; M0595, M0590, M0591, M0592, M0593, M0594, M0595, M0596, M0596, M0597, M0598, M0599; M060, M0600, M0601, M0602, M0603, M0604, M0605, M0607, M0608, M0609; M068, M0680, M0681, M0682, M0684, M0685, M0686, M0687, M0688, M0689; M069, M0690, M0691, M0692, M0693, M0694, M0695, M0696, M0697, M0698, M0699; M080, M0800, M0802, M0804, M0805, M0806, M0808, M0809

	41203	71400, 71401, 71402, 71403, 71404, 71405, 71406, 71407, 71408, 71409, 71423, 71424
Ulcerative colitis	20002	1463
	41270	K51, K510, K511, K512, K513, K514, K515, K518, K519; M075, M0750, M0751, M0755, M0758, M0759, M092
Multiple sclerosis	20002	1261
	41270	G35
	41203	3409
Crohn's disease	20002	1462
	41270	K50, K500, K501, K508, K509, M074, M0740, M0741, M0742, M0743, M0744, M0745, M0746, M0747, M0748, M0749, M091, M0910, M0911, M0912, M0913, M0914, M0915, M0916, M0917, M0918, M0919
Hyperthyroidism	20002	1225
	41270	E05, E050, E051, E052, E053, E054, E055, E058, E059; E062
	41203	2424, 2428, 2429, 7753
Lupus erythematosus	20002	1381
	41270	L93, L930, L931, L932; M32, M320, M321, M328, M329, M3290
	41203	6954, 7100
Age	21003	\
Alcohol consumption	20117	\
Body mass index	21001	\
Bone mineral density	4105	\
Education	6138	\
Falls	2296	\
Sex	31	\
Smoking	20116	\
Physical activity	884, 894, 904, 914	\
Depression	20002	1286, 1291, 1531
	41203	2961, 3004, 3119
	41270	F204, F313, F314, F315
Cognitive function	20002	1263
	41270	F067, R410, R411, R412, R413, R418

The use of sleep medicine	20003	1141151376, 1141194948, 1140917062, 1140863202, 1140855914, 1140865016, 1141171404, 1140863144, 1140862966, 1140862968, 1140876342, 1140855920, 1140863302, 1141171410, 1140879634, 1140867948, 1140879616, 1140867938, 1140867658, 1141168396, 1140867640, 1140882312, 1141169026, 1140855562
The use of glucocorticoid	20003	1140874816, 1141167174, 1141174548, 1140874790, 1140882622, 1140882764, 1140882766, 1140882774, 1140882780, 1141179982, 1140868364, 1140874930, 1140874976, 1140883026, 1141157402, 1140874896, 1140876456, 1140878562, 1140879922, 1140879934, 1140882822, 1140882824, 1140882830, 1140882836, 1140882840, 1140882842, 1140882844, 1140882846, 1140882848, 1140882850, 1140882864, 1140882888, 1140882894, 1140882896, 1140882902, 1140882906, 1140882908, 1140882914, 1140882918, 1140884672, 1140884704, 1140888134, 1140910424, 1140910634, 1141157294, 1141173346, 1141181062, 1141181610, 1141189464, 1141194840

It's known that the mechanisms/risk factors for the site of fracture (e.g. vertebral vs. non-vertebral) are different, grouping fracture by the site can tell this difference and also make your study comparable with previous studies.

Response: Thank you for your suggestions. Based on questionnaire-based self-reported fractures or hospital-based fractures using the ICD-9 or ICD-10 codes (Supplementary Table 4 or table below),

We identified 332 incident vertebral fractures in the UK Biobank dataset, this number was less than the number in another observational study using the UK Biobank,(5) this is because we excluded individuals who had fracture at baseline, when we included these samples, the number of vertebral fractures is similar with Luo et al.(5) After adjusting potential confounders (i.e., age, sex, BMI, smoking, alcohol consumption, physical activity, education, cognitive impairment, depression, and the use of glucocorticoid, benzodiazepines, and antidepressants), we found that participants with higher sleep risk scores (i.e., poor sleep quality) have increased risks of both vertebral and non-vertebral fracture (HR: 1.152, 95% CI: 1.011-1.313, *P*-value = 0.033 for vertebral fracture; HR: 1.086, 95% CI: 1.063-1.110, *P*-value = 7.71×10^{-14} for non-vertebral fracture). Accordingly, we have added this information in the result part of our manuscript on Pages 14, Lines 376-380. The changes were highlighted in red.

Detailed information on the UK Biobank field codes for fracture

Field ID	Code
20002	1626, 1627, 1628, 1629, 1630, 1631, 1632, 1633, 1634, 1635, 1636, 1637, 1638, 1639, 1640, 1644, 1645, 1646, 1647, 1648, 1649, 1650, 1651, 1652, 1653, 1654, 1655, 1656
41270	M484, M4840 , M4842, M4844, M4845, M4846 , M4847, M4848, M4849; M84, M840, M8400, M8401, M8402, M8403, M8404, M8405, M8406, M8407, M8408, M841, M8410, M8411, M8412, M8413, M8414, M8415, M8416, M8417, M8418, M8419, M842, M8421, M8422, M8423, M8424, M8425, M8426, M8427, M8428, M843, M8430, M8435, M8436, M8437, M8438, M8439, S02, S020, S0200, S0201, S021, S0210, S0211, S022, S0220, S0211, S023, S0230, S0231, S024, S0240, S0241, S025, S0250, S0251, S026, S0260, S0261, S027, S0270, S0271, S028, S0280, S0281, S029, S0290, S0291, S12, S120, S1200, S121, S1210, S122, S1220, S127, S1270, S128, S1280, S1281, S129, S1290, S1291, S22, S220, S2200, S221, S2210, S222, S2220, S223, S2230, S2231, S224, S2241, S225, S2250, S2251, S228, S2280, S229, S2290, S32, S320 , S3200 , S321, S3210, S3211, S322, S3220, S323, S3230, S3231, S324, S3240, S3241, S325, S3250, S3251, S327, S3270, S3271, S328, S3280, S3281, S42, S420, S4200, S4201, S421, S4210, S422, S4220, S4221, S423, S4230, S4231, S424, S4240, S4241, S427, S4270, S4271, S428, S4280, S429, S4290, S52, S520, S5200, S5201, S521, S5210, S5211, S522, S5220, S5221, S523, S5230, S5231, S524, S5240, S5241, S525, S5250, S5251, S526, S5260, S5261, S527, S5270, S5271, S528, S5280, S5281, S529, S5290, S62, S620, S6200, S621, S6210, S6211, S622, S6220, S6221, S623, S6230, S6231, S624, S6240, S6241, S625, S6250, S6251, S626, S6261, S627, S6270, S6271, S628, S6280, S6281, S72, S720, S7200, S7201, S721, S7210, S7211, S722, S7220, S7221, S723, S72230, S7231, S724, S7240, S7241, S727, S7270, S728, S7280, S729, S7290, S7291, S82, S820, S8200, S8201, S821, S8210, S8211, S822, S8220, S8221, S823, S8231, S824, S8240, S8241, S825, S8250, S8251, S826, S8260, S8261, S827, S8270, S8271, S828, S8280, S8281, S8286, S829, S8290, S8291, S92, S920, S9200, S9201, S921, S9210, S9211, S922, S9220, S9221, S923, S9230, S9231, S924, S9240, S9241, S925, S9250, S9251, S927, S9270,

S9271, S929, S9290, S9291; T02, T020, T0200, T021, T0210, T022, T0220, T023, T0230, T0231, T024, T0240, T0241, T025, T0250, T0251, T026, T0260, T027, T0270, T028, T0280, T029, T0290; **T08, T08X0**, T10, T10X0, T12, T142, T1420; T902, T911, T912, T921, T922, T931, T932, X5909, Z544
41203 7338, 73381, 73382, 73383, 73384, 73385, 73386, 73387, 73388, 73389; 800, 8000, 8001, 8002, 8003, 801, 8010, 8011, 802, 8020, 8022, 8023, 8024, 8026, 8028, 803, 8030, 8031; 805, 8050, 8052, **8054**, 8056, 8058; **806, 8064**, 807, 8070, 8072, 8074, 808, 8080, 8082, 8084, 8088, 8089, 809, 8090, 8091; 810, 8100, 811, 8110, 812, 8120, 8121, 8122, 8123, 8124, 8125, 813, 8130, 8131, 8132, 8134, 8135, 814, 8140, 8141, 815, 8150, 8151, 816, 8160, 8161, 817, 8170; 820, 8200, 8202, 8208, 8210, 8211, 8212; 822, 8220, 8221; 823, 8230, 8231, 8232, 8233, 8240, 8241, 8242, 8244, 8245, 8246, 8247, 8248, 8249, 825, 8250, 8252, 8253, 826, 8260, 8261, 828, 8280, 829, 8290; 905, 9050, 9052, 9053, 9054

The bold code means vertebral fracture

Did the authors adjust for baseline fracture? Those with fractures at baseline are more likely to have fractures in the future.

Response: This study is a prospective design, the participants who had suffered fractures before baseline were excluded, which can minimize the potential bias. We have stated this in the Method on Page 5, Line 107-108.

Discussion about why the results were different with different sleep behaviours is needed.

Response: Thank you for your suggestion. In fact, consistent results were observed for the 4 sleep traits (insomnia, sleep duration, excessive daytime sleepiness, and chronotype) in the observational study, while snoring was a protective factor for fracture risk.

We found that participants with insomnia showed a 7.0% higher risk of fracture. A U-shape relationship was observed between sleep duration and fracture risk ($P < 0.001$) with the lowest fracture risk at sleeping 7-8 hours per day. The excessive daytime sleepiness and “evening” chronotype were associated with fracture risk in the observational study. This is because sleep behaviors are often interconnected.(6) For example, participants with insomnia often have a shorter duration of sleep and excessive daytime sleepiness, and those with late chronotype often have reduced sleep duration. Hence, these sleep behaviors might individually act through different mechanisms, but could work synergistically to increase the risk of fracture.

In our study, we found that snoring was a protective factor for fracture risk, which was different from the previous 4 traits. While digging into our data, we found that participants with snoring were less likely to have insomnia and unnormal sleep duration (Supplementary Table 13 or table below), further analyses stratified by insomnia symptoms and sleep duration suggested that in the participants free of insomnia and with normal sleep duration, the association between snoring and

fracture risk was not statistically significant (HR: 1.063, 95% CI: 0.965-1.171, $P = 0.218$).

We have added discussion about this point in the Discussion section from Pages 15-18. The changes made were highlighted in red.

Baseline characteristics of participants with snoring and non-snoring.

Baseline characteristics	Snoring	
	No	Yes
Number of participants	205957	121621
Sleep duration ^a	7.16 (1.06)	7.21 (1.06)
Chronotype ^b		
Morning preference	188562 (91.6)	110545 (90.9)
Evening preference	17395 (8.4)	11076 (9.1)
Insomnia ^b		
No	48652 (23.6)	32262 (26.5)
Yes	157305 (76.4)	89359 (73.5)
Excessive daytime sleepiness ^b		
No	165130 (80.2)	89018 (73.2)
Yes	40827 (19.8)	32603 (26.8)

^a values are mean (SD); ^b values are numbers (percentages).

Discussion about potential mechanisms for sleep-fracture associations is also needed.

Response: As you suggested, we have added the discussion about the potential mechanism.

Insomnia and unnormal sleep duration were relevant to the metabolic disruption (e.g., decreased melatonin secretion)(7), and activated inflammation,(8) which might increase the risk of fracture.(9, 10) Late chronotype and short sleep duration have been linked to disrupted circadian rhythm,(11, 12) and the deficiency of clock gene (e.g., BMAL1) in mice could result in a low bone mass.(13) Additionally, circadian disruption could also increase sleepiness and possibly adversely affect vigilance to environmental hazards,(14) and increase the loss of postural control,(3) which result in the increased risks of falls.(15) And in our study, we observed that a larger proportion of the effect of sleep on fracture risk was mediated by falls than by BMD (e.g., 24.6% vs 10.7% for insomnia).

Please see Page 17, Lines 463 to 490. The changes were highlighted in red.

Reviewer #2 (Remarks to the Author):

The authors conducted a prospective study to infer the causality between sleep behaviors and fracture risk. Thanks to all the authors for this interesting manuscript in this important area. I suggest a minor revision for this manuscript. The comments are shown as follows:

1. Title

Please add the type of article in the title, I suggest "Observational and genetic evidence on the association of sleep behaviors with the incidence of fracture: A prospective study."

Response: Thank you for your suggestion. We did not add "A prospective study" in the title because we reported both observational and Mendelian randomization (MR) results. And the study design was described in both abstract and main text.

2. Abstract

Line 17: Please note the whole worlds of BMD for the first time in the manuscript;

Response: Thank you for your careful reading. The changes we made in the Abstract were highlighted in red.

3. DISCUSSION

Please clearly describe the LIMITATION of this study in the discussion.

Response: Thanks for your suggestion. Our study had a few limitations. First of all, the population included in this study were exclusively European and of a modest age range, which may limit the generalizability of our findings to other races and younger populations. Additionally, due to the sample overlap across some datasets used in the one-sample and two-sample MR analyses (e.g., snoring and sleep duration), one sample MR results could not be considered as independent replication of the two-sample MR results. We have added these limitations in the Discussion in the manuscript, which was highlighted in red (Lines 492-598 on Page 18).

4. OVERALL

The paper presents a topic of interest to the researchers in the related areas but the paper needs some language improvement before acceptance for publication.

Response: As you can see that we have made comprehensive changes according to the comments from reviewers, we have proofread the manuscript, and polished English writing.

Reviewer #3 (Remarks to the Author):

In this publication the authors triangulated various sources of statistical evidence to investigate the relationship between sleep traits and fracture risk, and concluded that an association is likely to exist between poor sleep quality and fracture risk. I enjoyed reading this paper- I found the research question interesting and the manuscript to be well written. I'm not an expert on statistical analysis involving longitudinal data so cannot comment on this in detail but I am well acquainted with two sample MR analysis. For these sections of the paper, the analysis was well conducted, and I support the conclusions that the authors make. I do however have some comments for improvements:

Major comments

Methods

- What p-value cut-off was used to select the instruments for the sleep traits, and how was the LD clumping conducted to ensure the instruments were independent? This detail is provided in the supplementary methods, but I think it is important and therefore would be better placed in the main manuscript.

Response: Thank you for your suggestion. In this study, for each sleep trait, we selected the genetic variants associated with corresponding phenotypes at a genome-wide significant level. After clumping for linkage disequilibrium at $r^2 < 0.01$ in 250kb regions, the genetic variants with the lowest *P*-value for the associated trait were included as final instrumental variables. We have added the detailed information in the Method section on Page 5 Lines 117-123. The changes we made were highlighted in red.

- For the same reason, I think details of the GWASs used for the exposures and outcomes tested in the two sample MR, which are currently reported in supplementary table 2, should be provided in the main text with citations to the relevant publications.

Response: Thank you for your suggestion. It has been done accordingly. The detailed information of the GWAS used in our studies was supplied and cited with the corresponding publications in the Method section of the manuscript (Lines 117-121 on Page 5 and line 131-134 on Page 6). The changes we made were highlighted in red.

- My understanding for the two sample MR study is that the authors used the sleep trait GWAS data from UK BioBank and then linked this to fracture risk measured in another study. I think clarification is needed on the sample overlap between the datasets used in the one-sample MR and two-sample MR study designs, and therefore whether these findings can be interpreted as replication from independent data sources or not.

Response: You are right. Most of the GWASs on exposures and outcome were conducted based on UK Biobank and 23andMe participants. The details on the sample source of GWAS datasets have been provided in the Supplementary Table 2 or table below. Due to the sample overlap between the datasets used in the one-sample and two-sample MR analyses, one sample MR results could not be considered as independent replication of the two-sample MR results. However, this would not change the final conclusion we made, because the MR results from two-sample and one-sample analyses were the same, and we combined conventional evidence from longitudinal data and genetic evidence from MR approach to infer the causality between sleep behaviors and fracture risk. We have discussed the corresponding contents in the Discussion part as the limitations of the study (Page 18, Line 495-498).

Detailed information on the genome-wide association studies for four sleep behaviors (chronotype, insomnia, snoring, and sleep duration) and summary statistics of fracture, falls, and bone mineral density.

Phenotype	Sample size	Sample source	PMID
Exposure			
Chronotype	697,828 individuals	UK Biobank and 23andMe participants	30696823
Insomnia	1,331,010 individuals	UK Biobank and 23andMe participants	30804565
Sleep duration	446,118 individuals	UK Biobank	30846698
Snoring	408,317 individuals (number of cases \approx 152,000; number of controls \approx 256,000)	UK Biobank	32060260
Outcome			
Bone mineral density	426,824 individuals	UK Biobank	30598549
Falls	361,194 individuals	UK Biobank	-
Fracture	426,795 individuals (number of cases = 53,184; number of controls = 373,611)	UK Biobank and 23andMe	30598549

- As there is a lot of different results presented in this paper, I think the reader would benefit from having the flow chart in supplementary figure 1 extended to summarize the main findings from each part of the study.

Response: Thank you for your suggestion. We moved the previous Supplementary Figure 1 to the main text as Figure 1, and added the main findings in this Figure.

Results

- F-statistics should be derived and reported to evaluate instrument strength for all the MR analyses.

Response: Thank you for your suggestion. Accordingly, we tested whether the sleep behavior-associated SNPs used as instrumental variables were appropriate instruments. We calculated the F-statistic for the SNP set used as instrumental variables using the methods as previously described by Burgess et al.(17) We found that for the instrumental variables for sleep-associated traits, all SNPs used greatly exceed the traditional strong instruments criteria, F-statistic ranging from 26.9 to 220.8. We have added the content of instrument strength in the Method section on Lines 126-129 Pages 5-6. The changes were highlighted in red.

- Q-statistics should also be derived and reported to evaluate instrument heterogeneity.

Response: Thank you for your suggestion. In our original analysis, we first performed the Cochran's

Q test to assess the heterogeneity among the effect estimates of the genetic variants used as instruments on each outcome. The Cochran's Q test suggested evidence of heterogeneity for each outcome (All P values < 0.05). Therefore, we used the inverse-variance weighted method based on a random-effects model to avoid overprecise estimates. We have reported the instrument heterogeneity in the Method section on Lines 234-235 on Page 9. The changes were highlighted in red.

- I think it would be informative to conduct a two sample MR-PheWas between the sleep trait instruments across fracture and bone-related risk factors and disease outcomes to see if any evidence of vertical pleiotropy can be detected, which would further support the causal relationship observed with fracture risk.

Response: Thank you for your suggestion. We have considered two main fracture-related risk factors: falls and BMD. We assessed the association between sleep behaviors and fall risks by two-sample MR. We found that genetically predicted insomnia and sleep duration were associated with fall risk (OR: 1.014, 95% CI: 1.006-1.022, $P = 0.001$ for insomnia; OR: 0.999, 95% CI: 0.999-0.999, $P < 0.001$ for sleep duration), supporting the hypothesis that falls might be a mediator for the sleep-fracture association, while the associations for the other two sleep behaviors were not significant (OR: 1.001, 95% CI: 0.993-1.009, $P = 0.861$ for chronotype; OR: 1.089, 95% CI: 0.974-1.217, $P < 0.001$ for snoring). We also assessed the associations of sleep behaviors with BMD by two-sample MR. However, evidence from the inverse-variance weighted analyses suggested that the associations were not statistically significant (OR: 1.015, 95% CI: 0.995-1.035, $P = 0.146$ for insomnia; OR: 1.000, 95% CI: 0.998-1.002, $P = 0.624$ for sleep duration; OR: 0.608, 95% CI: 0.364-1.016, $P = 0.057$ for snoring; OR: 1.009, 95% CI: 0.982-1.037, $P = 0.534$ for chronotype).

We have included these analyses in the Results. Please see Page 11, Lines 293-297, and Page 12, Lines 323-326.

- Steiger analysis or bi-directional MR should be performed to confirm correct orientation of causal effect.

Response: Thank you for your suggestion. Based on GWAS of 53,184 fracture cases and 373,611 controls, we included a total of 14 genetic variants as instruments for fracture, after clumping for linkage disequilibrium at $r^2 < 0.01$ in 250kb regions that reached genome-wide significance ($P < 5 \times 10^{-8}$). Summary-level data for sleep behaviors were obtained from GWAS in the UK Biobank.(21) Evidence from the inverse-variance-weighted method (under random-effects models) did not suggest the association between fracture and sleep behaviors (OR: 1.009, 95% CI: 0.968-1.051, $P = 0.660$ for chronotype; OR: 0.961, 95% CI: 0.853-1.083, $P = 0.514$ for snoring; OR: 1.000, 95% CI: 0.967-1.034, $P = 0.992$ for sleep duration; OR: 0.960, 95% CI: 0.896-1.028, $P = 0.244$ for insomnia), supporting the causal role of poor sleep quality on increased risk of fracture. We have added the result of bi-directional MR analyses in the result section of our manuscript on Lines 297-299 on Page 11 and Lines 326-328 on Page 12. The changes were highlighted in red.

Minor comments

These are recommendations for further analyses that might be useful and strengthen the conclusions of the paper. However, each analysis may require a lot of work so I don't think necessary to put in the manuscript, but the authors may want to run and include some of these.

- Any fracture was used as an outcome in this study. Would it be possible to repeat analyses using fracture data for the different sites as outcomes, and compare findings? A fracture site specific analysis might provide better power to detect sleep related effects and potentially help to further understand the causal mechanism (could be a discussion point in the manuscript).

Response: Thank you for your suggestion. We have extracted vertebral fractures based on questionnaire-based self-reported fractures or hospital-based fractures using the ICD-9 or ICD-10 codes from UK Biobank (Supplementary Table 4 or table below), we identified 332 incident vertebral fractures, this number was less than the number in another observational study using the UK Biobank,(5) this is because we excluded individuals who had a fracture at baseline, when we included these samples, the number of vertebral fractures is similar with Luo et al.(5). Based on the multivariable Cox regression model (i.e., adjusting age, sex, BMI, smoking, alcohol consumption, physical activity, education, cognitive impairment, depression, and the use of glucocorticoid, benzodiazepines, and antidepressants), we found that participants with higher sleep risk scores (i.e., poor sleep quality) have increased risks of both vertebral and non-vertebral fracture (HR: 1.152, 95% CI: 1.011-1.313, *P*-value = 0.033 for vertebral fracture; HR: 1.086, 95% CI: 1.063-1.110, *P*-value = 7.71×10^{-14} for non-vertebral fracture). Accordingly, we have added this information in the result part of our manuscript on Page 14, Lines 377-380. The changes were highlighted in red.

Detailed information on the UK Biobank field codes for fracture

Field ID	Code
20002	1626, 1627, 1628, 1629, 1630, 1631, 1632, 1633, 1634, 1635, 1636, 1637, 1638, 1639, 1640, 1644, 1645, 1646, 1647, 1648, 1649, 1650, 1651, 1652, 1653, 1654, 1655, 1656
41270	M484, M4840 , M4842, M4844, M4845, M4846 , M4847, M4848, M4849; M84, M840, M8400, M8401, M8402, M8403, M8404, M8405, M8406, M8407, M8408, M841, M8410, M8411, M8412, M8413, M8414, M8415, M8416, M8417, M8418, M8419, M842, M8421, M8422, M8423, M8424, M8425, M8426, M8427, M8428, M843, M8430, M8435, M8436, M8437, M8438, M8439, S02, S020, S0200, S0201, S021, S0210, S0211, S022, S0220, S0211, S023, S0230, S0231, S024, S0240, S0241, S025, S0250, S0251, S026, S0260, S0261, S027, S0270, S0271, S028, S0280, S0281, S029, S0290, S0291, S12, S120, S1200, S121, S1210, S122, S1220, S127, S1270, S128, S1280, S1281, S129, S1290, S1291, S22, S220, S2200, S221, S2210, S222, S2220, S223, S2230, S2231, S224, S2241, S225, S2250, S2251, S228, S2280, S229, S2290, S32, S320 , S3200 , S321, S3210, S3211, S322, S3220, S323, S3230, S3231, S324, S3240, S3241, S325, S3250, S3251, S327, S3270, S3271, S328, S3280, S3281, S42, S420, S4200, S4201, S421, S4210, S422, S4220, S4221, S423, S4230, S4231, S424, S4240, S4241, S427, S4270, S4271, S428, S4280, S429,

S4290, S52, S520, S5200, S5201, S521, S5210, S5211, S522, S5220, S5221, S523, S5230, S5231, S524, S5240, S5241, S525, S5250, S5251, S526, S5260, S5261, S527, S5270, S5271, S528, S5280, S5281, S529, S5290, S62, S620, S6200, S621, S6210, S6211, S622, S6220, S6221, S623, S6230, S6231, S624, S6240, S6241, S625, S6250, S6251, S626, S6261, S627, S6270, S6271, S628, S6280, S6281, S72, S720, S7200, S7201, S721, S7210, S7211, S722, S7220, S7221, S723, S72230, S7231, S724, S7240, S7241, S727, S7270, S728, S7280, S729, S7290, S7291, S82, S820, S8200, S8201, S821, S8210, S8211, S822, S8220, S8221, S823, S8231, S824, S8240, S8241, S825, S8250, S8251, S826, S8260, S8261, S827, S8270, S8271, S828, S8280, S8281, S8286, S829, S8290, S8291, S92, S920, S9200, S9201, S921, S9210, S9211, S922, S9220, S9221, S923, S9230, S9231, S924, S9240, S9241, S925, S9250, S9251, S927, S9270, S9271, S929, S9290, S9291; T02, T020, T0200, T021, T0210, T022, T0220, T023, T0230, T0231, T024, T0240, T0241, T025, T0250, T0251, T026, T0260, T027, T0270, T028, T0280, T029, T0290; **T08, T08X0**, T10, T10X0, T12, T142, T1420; T902, T911, T912, T921, T922, T931, T932, X5909, Z544

41203

7338, 73381, 73382, 73383, 73384, 73385, 73386, 73387, 73388, 73389; 800, 8000, 8001, 8002, 8003, 801, 8010, 8011, 802, 8020, 8022, 8023, 8024, 8026, 8028, 803, 8030, 8031; 805, 8050, 8052, **8054**, 8056, 8058; **806, 8064**, 807, 8070, 8072, 8074, 808, 8080, 8082, 8084, 8088, 8089, 809, 8090, 8091; 810, 8100, 811, 8110, 812, 8120, 8121, 8122, 8123, 8124, 8125, 813, 8130, 8131, 8132, 8134, 8135, 814, 8140, 8141, 815, 8150, 8151, 816, 8160, 8161, 817, 8170; 820, 8200, 8202, 8208, 8210, 8211, 8212; 822, 8220, 8221; 823, 8230, 8231, 8232, 8233, 8240, 8241, 8242, 8244, 8245, 8246, 8247, 8248, 8249, 825, 8250, 8252, 8253, 826, 8260, 8261, 828, 8280, 829, 8290; 905, 9050, 9052, 9053, 9054

The bold code means vertebral fracture

- Including multivariable MR analysis (MVMR) as follow-up for the two sample MR, where the sleep trait exposures are modelled together to find the independent MR effects on risk fracture could be informative.

Response: Thank you for your suggestion. After clumping for linkage disequilibrium at $r^2 < 0.01$ in 250kb regions, genetic variants associated with sleep behavior (chronotype,(22) insomnia,(21) sleep duration,(23) or snoring(24)) at a genome-wide significant ($P < 5 \times 10^{-8}$) level were selected from the largest genome-wide association studies. We constructed novel instrumental variables by the combination of SNPs and their genetic association estimates from these GWASs. The genetic association estimates of these instrumental variables with other exposures (i.e., other sleep-related factors, except for the originally associated sleep behavior) and outcome (i.e., fracture) were obtained from the summary statistics from GWAS in the UK Biobank.(19, 21) Accordingly, using MendelianRandomization packages in R software, we conducted the multivariable MR analyses, where insomnia, sleep duration, snoring, and chronotypes were modeled. We found that after controlling for sleep duration, snoring, and chronotype, the association between insomnia and

fracture risk remained significant (OR: 1.055; 95% CI: 1.027-1.085, $P = 1.36 \times 10^{-4}$). Additionally, we also observed a significant association for sleep duration, when controlling for other sleep behaviors (OR: 0.997, 95% CI: 0.995-0.999, $P = 0.011$). For snoring and chronotype, the associations were not significant (OR: 0.994, 95% CI: 0.960-1.030, $P = 0.715$ for chronotype; (OR: 1.062, 95% CI: 0.976-1.155, $P = 0.161$ for snoring). These data, taken together, suggested independent causal effects of insomnia and sleep duration on fracture risks. We have revised the Method and Result in the manuscript (Page 10, Line 250-258, and Pages 13-14, Line 356-362). The changes were highlighted in red.

- For completeness, it might be worthwhile conducting an MR analysis between your sleep risk score and fracture risk.

Response: Thank you for your suggestion. Due to the lack of a genome-wide association study for sleep risk score, we were unable to conduct the Mendelian randomization analyses of the association between sleep risk score and the risk of fracture.

Reference

1. Cawthon PM. Gender differences in osteoporosis and fractures. *Clin Orthop Relat Res* 2011;469:1900-1905.
2. Trajanoska K, Morris JA, Oei L, Zheng HF, Evans DM, Kiel DP, et al. Assessment of the genetic and clinical determinants of fracture risk: genome wide association and mendelian randomisation study. *BMJ* 2018;362:k3225.
3. Robillard R, Prince F, Boissonneault M, Filipini D, Carrier J. Effects of increased homeostatic sleep pressure on postural control and their modulation by attentional resources. *Clin Neurophysiol* 2011;122:1771-1778.
4. Stone KL, Blackwell TL, Ancoli-Israel S, Cauley JA, Redline S, Marshall LM, et al. Sleep disturbances and risk of falls in older community-dwelling men: the outcomes of Sleep Disorders in Older Men (MrOS Sleep) Study. *J Am Geriatr Soc* 2014;62:299-305.
5. Luo J, Lee RY. How Does Obesity Influence the Risk of Vertebral Fracture? Findings From the UK Biobank Participants. *JBMR Plus* 2020;4:e10358.
6. Fan M, Sun D, Zhou T, Heianza Y, Lv J, Li L, et al. Sleep patterns, genetic susceptibility, and incident cardiovascular disease: a prospective study of 385 292 UK biobank participants. *Eur Heart J* 2020;41:1182-1189.
7. Blask DE. Melatonin, sleep disturbance and cancer risk. *Sleep Med Rev* 2009;13:257-264.
8. Grandner MA, Buxton OM, Jackson N, Sands-Lincoln M, Pandey A, Jean-Louis G. Extreme sleep durations and increased C-reactive protein: effects of sex and ethnorracial group. *Sleep* 2013;36:769-779E.
9. Frisher M, Gibbons N, Bashford J, Chapman S, Weich S. Melatonin, hypnotics and their association with fracture: a matched cohort study. *Age Ageing* 2016;45:801-806.
10. Ishii S, Cauley JA, Greendale GA, Crandall CJ, Danielson ME, Ouchi Y, et al. C-reactive protein, bone strength, and nine-year fracture risk: data from the Study of Women's Health Across the Nation (SWAN). *J Bone Miner Res* 2013;28:1688-1698.
11. Horne JA, Ostberg O. A self-assessment questionnaire to determine morningness-eveningness in human circadian rhythms. *Int J Chronobiol* 1976;4:97-110.

12. Yu JH, Yun CH, Ahn JH, Suh S, Cho HJ, Lee SK, et al. Evening chronotype is associated with metabolic disorders and body composition in middle-aged adults. *J Clin Endocrinol Metab* 2015;100:1494-1502.
13. Samsa WE, Vasanthi A, Midura RJ, Kondratov RV. Deficiency of circadian clock protein BMAL1 in mice results in a low bone mass phenotype. *Bone* 2016;84:194-203.
14. Garbarino S, Mascialino B, Penco MA, Squarcia S, De Carli F, Nobili L, et al. Professional shift-work drivers who adopt prophylactic naps can reduce the risk of car accidents during night work. *Sleep* 2004;27:1295-1302.
15. Stone KL, Ewing SK, Lui LY, Ensrud KE, Ancoli-Israel S, Bauer DC, et al. Self-reported sleep and nap habits and risk of falls and fractures in older women: the study of osteoporotic fractures. *J Am Geriatr Soc* 2006;54:1177-1183.
16. Swanson CM, Shea SA, Stone KL, Cauley JA, Rosen CJ, Redline S, et al. Obstructive sleep apnea and metabolic bone disease: insights into the relationship between bone and sleep. *J Bone Miner Res* 2015;30:199-211.
17. Burgess S, Thompson SG, Collaboration CCG. Avoiding bias from weak instruments in Mendelian randomization studies. *Int J Epidemiol* 2011;40:755-764.
18. Kanis JA. Diagnosis of osteoporosis and assessment of fracture risk. *Lancet* 2002;359:1929-1936.
19. Morris JA, Kemp JP, Youten SE, Laurent L, Logan JG, Chai RC, et al. An atlas of genetic influences on osteoporosis in humans and mice. *Nat Genet* 2019;51:258-266.
20. Edwards MH, Jameson K, Denison H, Harvey NC, Sayer AA, Dennison EM, et al. Clinical risk factors, bone density and fall history in the prediction of incident fracture among men and women. *Bone* 2013;52:541-547.
21. Jansen PR, Watanabe K, Stringer S, Skene N, Bryois J, Hammerschlag AR, et al. Genome-wide analysis of insomnia in 1,331,010 individuals identifies new risk loci and functional pathways. *Nat Genet* 2019;51:394-403.
22. Jones SE, Lane JM, Wood AR, van Hees VT, Tyrrell J, Beaumont RN, et al. Genome-wide association analyses of chronotype in 697,828 individuals provides insights into circadian rhythms. *Nature communications* 2019;10:343.
23. Dashti HS, Jones SE, Wood AR, Lane JM, van Hees VT, Wang H, et al. Genome-wide association study identifies genetic loci for self-reported habitual sleep duration supported by accelerometer-derived estimates. *Nature communications* 2019;10:1100.
24. Campos AI, Garcia-Marin LM, Byrne EM, Martin NG, Cuellar-Partida G, Renteria ME. Insights into the aetiology of snoring from observational and genetic investigations in the UK Biobank. *Nature communications* 2020;11:817.

REVIEWERS' COMMENTS:

Reviewer #1 (Remarks to the Author):

The authors have addressed my concerns.

Reviewer #2 (Remarks to the Author):

Thanks for the revisions. Most of the comments have been clearly addressed, which have significantly improved the paper.

Reviewer #3 (Remarks to the Author):

Well done on completing all the corrections. I'm satisfied with the authors response to my comments and amendments made to the manuscript. There is one minor change I would suggest. To address my MR-PheWas comment the authors looked at BMD and falls outcome which although is reasonable it would have been better to extend this to look at a range of traits to provide a fuller view of the pleiotropic landscape. For example, to see if the MR-PheWas detects a relationship with a confounder of sleep versus fracture risk indicating the MR estimate could be biased. In addition, it would also be interesting to know if sleep scores are related to any bone related diseases.

Title: Observational and genetic evidence on the association of sleep behaviors with the incidence of fracture

Thank you for the opportunity to resubmit our manuscript to *Communications Biology* for consideration for publication. For ease of reading we have directly pasted the Reviewer's comments below, which are in Calibri font.

Our responses are in Times New Roman font, preceded by “**Response:**”

Reviewers' comments:

Reviewer #1 (Remarks to the Author):
The authors have addressed my concerns.

Response: Thank you so much.

Reviewer #2 (Remarks to the Author):
Thanks for the revisions. Most of the comments have been clearly addressed, which have significantly improved the paper.

Response: Thank you so much.

Reviewer #3 (Remarks to the Author):
Well done on completing all the corrections. I'm satisfied with the authors response to my comments and amendments made to the manuscript. There is one minor change I would suggest. To address my MR-PheWas comment the authors looked at BMD and falls outcome which although is reasonable it would have been better to extend this to look at a range of traits to provide a fuller view of the pleiotropic landscape. For example, to see if the MR-PheWas detects a relationship with a confounder of sleep versus fracture risk indicating the MR estimate could be biased. In addition, it would also be interesting to know if sleep scores are related to any bone related diseases.

Response: Thank you for your suggestion. MR-PheWAS analysis has been widely used as a *hypothesis-free* approach with the aims of assessing causal associations of an exposure with a wide range of outcomes [1]. MR-PheWAS results could provide pleiotropic landscape of sleep traits, however, the aim of this study is to assess the causal relationships between sleep behaviors and fracture risk. In this study design, we have considered the potential confounders of sleep versus fracture risk (*under hypothesis*), we excluded participants with potential comorbid diseases (i.e., rheumatoid arthritis, ulcerative colitis, multiple sclerosis, Crohn's disease, hyperthyroidism,

or lupus erythematosus). In the multivariable Cox regression, the basic model was adjusted for confounders, including age, sex, body mass index, education, smoking, alcohol consumption, physical activity, cognitive impairment, depression, and the use of glucocorticoid medication, benzodiazepines, and antidepressants. Additional covariables such as BMD and falls were also included to explore the potential role of falls and BMD on the associations between sleep behaviors and fracture risk.

We have assessed the causality between sleep behaviors and falls and BMD (the two main risk factors for fracture) by two-sample MR. As you requested, we further included several risk factors (e.g., body mass index, and grip strength), and explored the causal associations between sleep behaviors and these factors. There was no significant evidence to support the association of genetically predicted chronotype and snoring with these factors (**Table below**). However, based on Bonferroni-correction, genetically predicted insomnia were statistically significant associated with muscle weakness [odds ratio (OR): 1.066, 95% confidence interval (95% CI): 1.027-1.107, $P = 0.001$ for muscle weakness]. Additionally, genetically predicted insomnia was inversely associated with body mass index (β -coefficient = -0.072, standard error = 0.019, $P = 1.81 \times 10^{-4}$). To assess the influence of these factors on MR estimates of relationships between insomnia and fracture risk, we included potential factors as covariates (i.e., age, sex, body mass index, education, smoking, alcohol consumption, physical activity, cognitive impairment, depression, and the use of glucocorticoid medication, benzodiazepines, and antidepressants) in one-sample MR. The association between insomnia and fracture was significant [hazard ratio (HR): 1.061, 95% CI: 1.003-1.123, $P = 0.040$]. MR-PheWAS analysis might bring more and more results (for example dozens of traits), if all results were included, the paper might be out of focus.

While MR-PheWAS could provide pleiotropic landscape of the trait, the MR estimate might be biased by the horizontal pleiotropic effect of instrumental variables, to assess the robustness of the MR estimate of the relationships of sleep behaviors with fracture risk, we assessed the presence of horizontal pleiotropy of instrumental variables by conducting the MR-Egger test and MR Pleiotropy Residual Sum and Outlier (MR-PRESSO) analyses. MR-Egger regression showed no evidence of directional pleiotropy (all P for intercept > 0.05). After excluding outliers ($N = 1$ for sleep duration; $N = 1$ for insomnia), the causal association estimates for insomnia and sleep duration remained significant (OR: 1.056, 95% CI: 1.027-1.086, $P = 1.94 \times 10^{-4}$ for insomnia; OR: 0.997, 95% CI: 0.995-0.999, $P = 0.999$ for sleep duration).

Additionally, for the association between sleep scores and bone-related diseases, we here analyzed rheumatoid arthritis (RA) and systemic lupus erythematosus (SLE) as outcomes. Based on the UK Biobank dataset, after excluding participants without European ancestry, 472,020 participants were included in this analysis. Among them, we identified 8895 RA and 852 SLE cases. Multivariable logistic regression was used to assess the associations of sleep risk score with RA and SLE. After adjusting for confounders (i.e., age, sex, body mass index, education, smoking, alcohol consumption, physical activity, cognitive impairment, depression, and the use of glucocorticoid medication, benzodiazepines, and antidepressants), the association for RA and SLE were statistically significant (OR: 1.243, 95% CI: 1.209-1.277, $P < 2 \times 10^{-16}$ for RA; OR: 1.337, 95% CI: 1.229-1.456, $P = 2.04 \times 10^{-11}$ for SLE). Hence, in this study, we have excluded participants

with these diseases, to minimize the influence of these diseases on the sleep-fracture association estimate.

Table the MR association between sleep behaviors and several fracture risk factors based on the IVW method

Trait	Outcome	beta/OR	se/95% CI	P
Chronotype	Body mass index	-0.066	0.037	0.075
Insomnia	Body mass index	-0.072	0.019	1.81×10 ⁻⁴
Sleep duration	Body mass index	0.004	0.004	0.394
Snoring	Body mass index	-0.234	0.448	0.602
Chronotype	Muscle weakness*	0.988	0.950-1.028	0.546
Insomnia	Muscle weakness*	1.066	1.027-1.107	0.001
Sleep duration	Muscle weakness*	0.997	0.995-0.999	0.024
Snoring	Muscle weakness*	1.289	0.753-2.206	0.354

* means that OR and 95% CI were used to describe this association; # means that beta and se were used to describe this association.

Abbreviations: IVW, inverse-variance weighted; MR, Mendelian randomization; OR, odds ratio; 95% CI, 95% confidence interval

Reference

1. Nicolopoulos, K., et al., *Association between habitual coffee consumption and multiple disease outcomes: A Mendelian randomisation phenome-wide association study in the UK Biobank*. Clin Nutr, 2020. **39**(11): p. 3467-3476.